# Lamprey VLRB response to influenza virus supports universal rules of immunogenicity and antigenicity

Meghan O Altman[1], Jack R Bennink[1], Jonathan W Yewdell[1]*, Brantley R Herrin[2]

[1]Laboratory of Viral Diseases, National Institute of Allergy and Infectious Diseases, Bethesda, United States; [2]Department of Pathology and Laboratory Medicine, Emory University School of Medicine, Atlanta, United States

**Abstract** Immunoglobulins (Igs) are a crown jewel of jawed vertebrate evolution. Through recombination and mutation of small numbers of genes, Igs can specifically recognize a vast variety of natural and man-made organic molecules. Jawless vertebrates evolved a parallel system of humoral immunity, which recognizes antigens not with Ig, but with a structurally unrelated receptor called the variable lymphocyte receptor B (VLRB). We exploited the convergent evolution of Ig and VLRB antibodies (Abs) to investigate if intrinsic chemical features of foreign proteins determine their antigenicity and immunogenicity. Surprisingly, we find lamprey VLRB and mouse Ig responses to influenza A virus are extremely similar. Each focuses ~80% of the response on hemagglutinin (HA), mainly through recognition of the major antigenic sites in the HA globular head domain. Our findings predict basic conservation of Ab responses to protein antigens, strongly supporting the use of animal models for understanding human Ab responses to viruses and protein immunogens.

*For correspondence: jyewdell@nih.gov

Competing interests: The authors declare that no competing interests exist.

## Introduction

A cornerstone of modern biology and medicine is that humans and other mammals generate a highly specific humoral immune response when confronted with microbes or toxins, providing the basis for vaccination against many pathogens. Originally defined as a functional principle (e.g., ability to protect animals against injection with a toxin), the responsible substance was termed an antibody (Ab), which we now know to consist of immunoglobulins (Igs) in jawed vertebrates. After more than a century of steady progress, a basic molecular understanding of Ig function is at hand (*Ramaraj et al., 2012*; *Georgiou et al., 2014*); surprisingly little is known, however, regarding the basic rules of immunogenicity. In responding to viruses, why are some proteins more immunogenic than others? Why do Ig responses focus on certain regions of proteins? To what extent is this due to the specific features of Ig structure and combining site chemistry or repertoire limitations imposed by self-tolerance?

Here we address these issues by comparing the Ab responses of lampreys and mice to influenza A virus (IAV), a model antigen of high practical relevance. IAV is composed of four major structural proteins: two surface glycoproteins, hemagglutinin (HA) and neuraminidase (NA), embedded in a lipid envelope, lined by the matrix protein (M1), which encases the nucleoprotein (NP) coated viral genome. Due to its importance as the target of protective Igs (*Couch and Kasel, 1983*), HA is probably the most intensively characterized immunogen/antigen. Most HA-specific Igs with virus neutralizing activity bind to or bridge 5 antigenic regions in the globular domain (termed Sa, Sb, Ca1, Ca2, and Cb), which surround the sialic acid receptor site that attaches HA to host cells (*Gerhard et al., 1981*). Variation in these sites as IAV evolves in the human population ('antigenic drift') prevents effective IAV vaccination, necessitating frequent

**eLife digest** Influenza viruses infect ten of millions of people each year. To conquer a flu infection, the human immune system develops antibodies that hasten recovery and prevent future flu infections. Unfortunately, flu is constantly changing in response to the human immune response, and antibodies induced by previous infection or vaccination provide partial protection, at best, against new strains.

An ideal flu vaccine would stimulate the immune system to produce antibodies that protect against all future strains of influenza. Most human antibodies that are induced by influenza target a part of the virus called the hemagglutinin, which attaches the virus to cells to start a flu infection. Some hemagglutinin-specific antibodies recognize many strains of influenza, but individuals do not produce enough of these antibodies to prevent infections with new strains. A basic understanding of what drives the production of different types of antibodies is important to devise vaccines that produce broadly effective antibodies for flu and for other viruses and pathogens that have proven to be difficult vaccine targets.

To better understand the rules of antibody generation, Altman et al. compared antibodies produced in response to flu in mice and lampreys. Lampreys are a primitive fish that branched off from other vertebrates (animals with a backbone, like you) 550 million years ago and developed their own system of antibody recognition based on a completely different template. Despite this, Altman et al. found that the antibody response of mice and lampreys to flu is remarkably similar. Of several potential viral targets, antibodies from both mice and lampreys were predominantly directed against the hemagglutinin. Of numerous potential locations on the hemagglutinin, mouse and lamprey antibodies predominantly recognized the same region.

These similarities suggest that the specificity of antibodies is based largely on the properties of the virus, and varies little with the properties of the responding organism. Most importantly, this supports the conclusion that studies in mice and other mammals are likely to accurately predict how humans will respond to vaccines for viruses and other pathogens.

changes in vaccine formulation. The recent discovery that humans can generate protective Igs to conserved structures in the membrane-proximal HA stem have raised hopes of more effective vaccination if stem responses can be augmented using appropriately designed vaccines (*Laursen and Wilson, 2013*). Better understanding the rules of immunogenicity could inform these critical efforts.

We characterized the variable lymphocyte receptor B (VLRB) response of lampreys, which along with hagfish, are the only known living jawless vertebrates (cyclostomes). Cyclostomes branched evolutionarily from jawed vertebrates approximately 550 million years ago (Mya) (*Figure 1*). Pioneering studies from the Good laboratory established that lampreys generate Ab responses to protein and carbohydrate immunogens (*Finstad and Good, 1964*). More than 40 years later, Cooper and colleagues discovered that lamprey Abs, VLRBs, bear no relationship to Igs, but rather are structurally similar to Toll-like receptors (*Pancer et al., 2004*). In place of RAG-mediated V(D)J recombination, VLRB diversity is generated using a gene assembly mechanism reminiscent of the activation-induced cytidine deamine-catalyzed gene conversion mechanism used to diversify *Ig* genes in birds and some mammals (*Figure 1*).

The germline *VLRB* gene is incomplete because the invariant 5′ and 3′ coding sequences are separated by non-coding intervening sequences (*Pancer et al., 2004*). Several of the hundreds of leucine rich repeat (LRR)-encoding genes flanking the *VRLB* gene are copied into the gene to generate an in-frame, functional *VLRB* gene during lymphocyte development (*Nagawa et al., 2007*; *Rogozin et al., 2007*; *Alder et al., 2008*). This generates a VLRB repertoire with diversity comparable to Igs (*Alder et al., 2005*). *VLRB* encodes for single-chain, crescent-shaped proteins that bind to antigens with a concave surface composed of multiple LRR β-strands and a C-terminal variable loop (LRRCT) (*Kim et al., 2007*; *Han et al., 2008*; *Herrin et al., 2008*; *Velikovsky et al., 2009*; *Kirchdoerfer et al., 2012*; *Deng et al., 2013*; *Luo et al., 2013*). In contrast, Igs consist of a heavy and a light chain, each of which contributes three complementarity determining region loops to form a structurally distinct antigen-binding site (*Figure 1*).

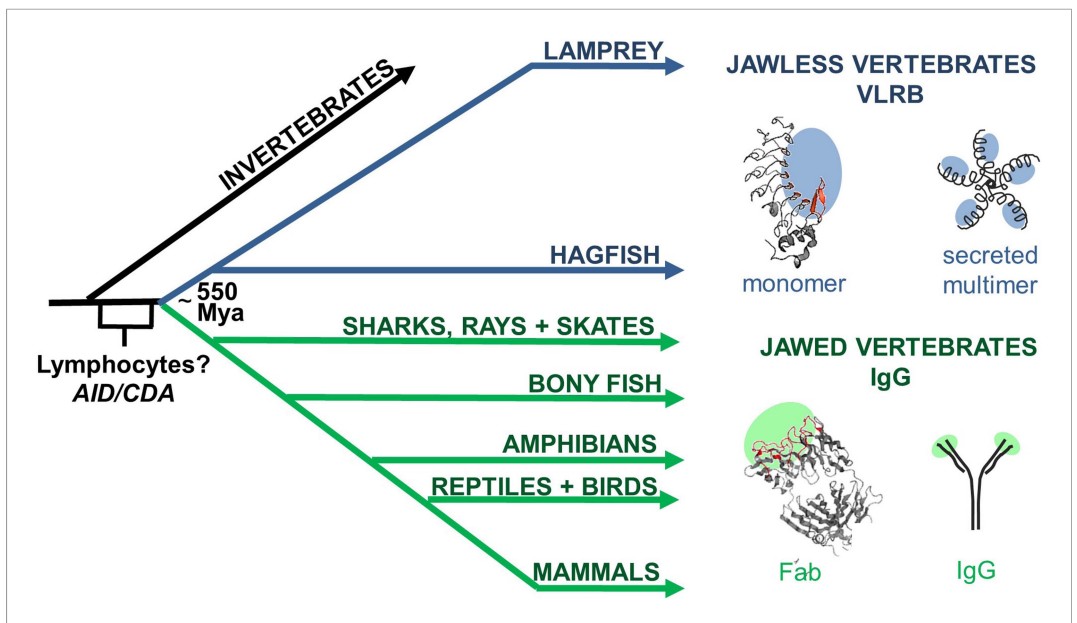

**Figure 1**. Origin of variable lymphocyte receptor B (VLRB) in jawless vertebrates. Jawless and jawed vertebrates last shared a common ancestor ~550 Mya. VLR genes are only in jawless vertebrates, whereas Immunoglobulin (Ig) genes are only in jawed vertebrates. However, both jawed and jawless vertebrates have a lymphocyte-based adaptive immune system suggesting that the genetic programs necessary for lymphocyte development originated in a common ancestor before the antigen receptor genes. Cytidine deaminases are expressed by lymphocytes in both jawed and jawless vertebrates and may have originated in a common ancestor; activation-induced cytidine deaminase (AID) and cytosine deaminase (CDA). Structures of prototypic VLRB (Top, PDB: 3e6j) and IgG (Bottom, PDB: 1Igt) are shown to the right, along with cartoons of their secreted forms. Regions of antigen recognition are shaded in blue or green. In red are the concave antigen binding residues of VLR and the complementarity determining regions (CDRs) of Ig.

## Results

To probe the VLRB response to IAV, we collected blood from lamprey larvae immunized three times with inactivated, purified prototypic H1N1 PR8 IAV. Polyclonal VLRB primarily migrates on an SDS-PAGE gel as disulfide-linked multimers under non-reducing conditions and as monomers in the presence of reducing agents (*Alder et al., 2008*; *Herrin et al., 2008*). As seen previously (*Alder et al., 2005*), monitoring plasma VLRB by immunoblotting revealed that unlike mammalian Ig, where immunization induces only minor increases in substantial serum levels, VLRB levels increase ~sevenfold (*Figure 2A*). ELISAs revealed that each immunized lamprey generated VLRBs that bind PR8, but not a similar amount of plate-bound parainfluenza-3 virus, which is a genetically and serologically completely distinct enveloped virus, but similar in architecture and complexity to IAV (*Figure 2B*).

To determine the immunogenicity of IAV structural proteins, we measured serum from PR8-immunized mice and lamprey via ELISA using either detergent soluble proteins from purified virus (HA, NA, M1), or the detergent insoluble core (NP, M1, small amounts of other non-glycoproteins [*Hutchinson et al., 2014*]) (*Figure 3A* and *Figure 3—figure supplement 1*). This revealed that in both mice and lamprey, more than 90% of the functional ELISA response is specific for HA and NA, as shown by the large difference in titers between detergent soluble proteins from PR8 (H1N1) vs X31 (H3N2), a reassortant virus with the PR8 internal proteins but serologically distinct HK68 glycoproteins. Genetically isolating HA from NA using the J1 (H3N1, PR8 internal proteins) and P50 (H1N2, HK internal proteins) reassortants shows that upwards of 80% of ELISA-detected Abs are specific for HA in lamprey and mice (*Figure 3B*). Low binding to X31 and HK soluble proteins, which contain significant amounts of M1 (*Figure 3—figure supplement 1*), indicate that M1 is negligibly immunogenic (note that internal viral proteins from H3 and H1 viruses are antigenically highly conserved). Further, the low serum titers against PR8 cores confirm that only a small fraction of Igs are specific for NP or a low abundance internal virion component.

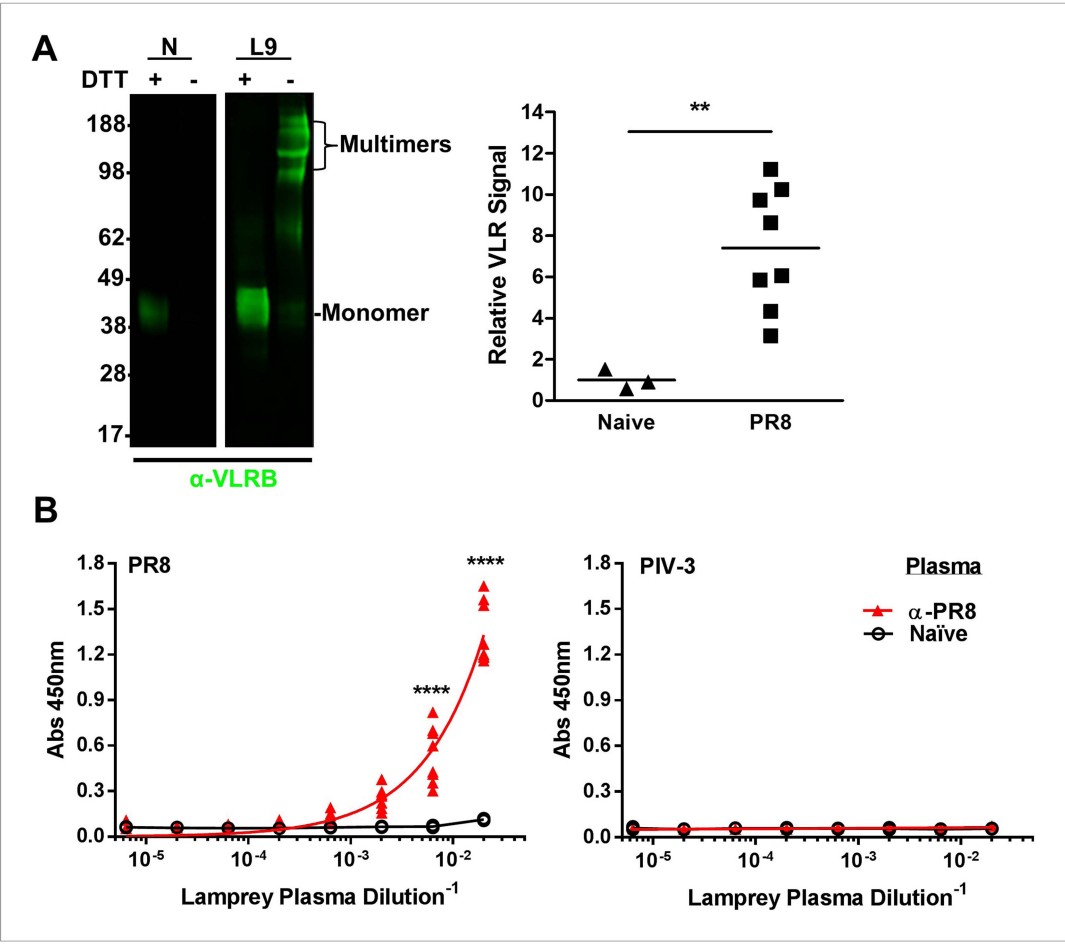

**Figure 2**. Lamprey make VLRBs specific for influenza A virus (IAV) after immunization with non-adjuvented, UV-inactivated virus. (**A**) Left, whole lamprey plasma (5 µl of naïve or immunized three times with PR8 [L9]) electrophoresed on a 4–12% SDS PAGE gel probed with anti-VLR monoclonal Ab (mAb) by immunoblotting. VLR monomers (~35–45 kDa) are naturally cross-linked by disulfide bonds to form VLR multimers >100 kDa. Right, lane intensity measured by ImageJ for immunoblots of 2 µl Naïve (3 animals) or PR8 immunized (8 animals) probed with anti-VLR 2° Ab. Each point represents one animal. Data were analyzed by two-tailed t-test using PRISM software (\*\*p < 0.01). The mean signal from immunized plasma was 7.4 ± 1.8 × greater than the naïve mean. (**B**) Equal protein quantities of purified virus were adsorbed to ELISA plates and probed with lamprey plasma from either immunized (n = 9) or naïve (n = 2) animals. Data were analyzed by two-way ANOVA followed by Bonferroni multiple comparison using PRISM software (\*\*\*\*p < 0.0001).

Reciprocal immunization of lampreys with HK virus (*Figure 3C*) confirmed the dominance of HA. This experiment also provides a direct control for the specificity of lamprey VLRB for H1N1 vs H3N2 glycoproteins. Flow cytometry of cells expressing either HA, NA, NP, M1/M2 or NS1 (which is present in virions [*Hutchinson et al., 2014*]) from transfected cDNAs stained with lamprey plasma showed that PR8 induced detectable VLRB responses to HA and NA but not NP, M1, M2, or NS1 (*Figure 3—figure supplement 2*). Similarly, mouse serum Ig was positive against HA and NA and negative for M1 + M2, although there was a response to NP. While lamprey plasma did not bind plasmid expressed NP by flow, in ELISA, both immune lamprey plasma and mouse sera bound plated NP, but neither bound M1 (*Figure 3—figure supplement 3*). The lack of NP binding in the flow assay is most likely spurious; due to limited VLRB access to NP within permeabilized cells, or low signal.

Next we examined the functionality of the lamprey anti-HA response as revealed by hemagglutination inhibition (HI) or infectivity neutralization assays. HI measures the ability of Abs to block HA-mediated IAV attachment to erythrocyte surface terminal sialic acids. PR8-immunized

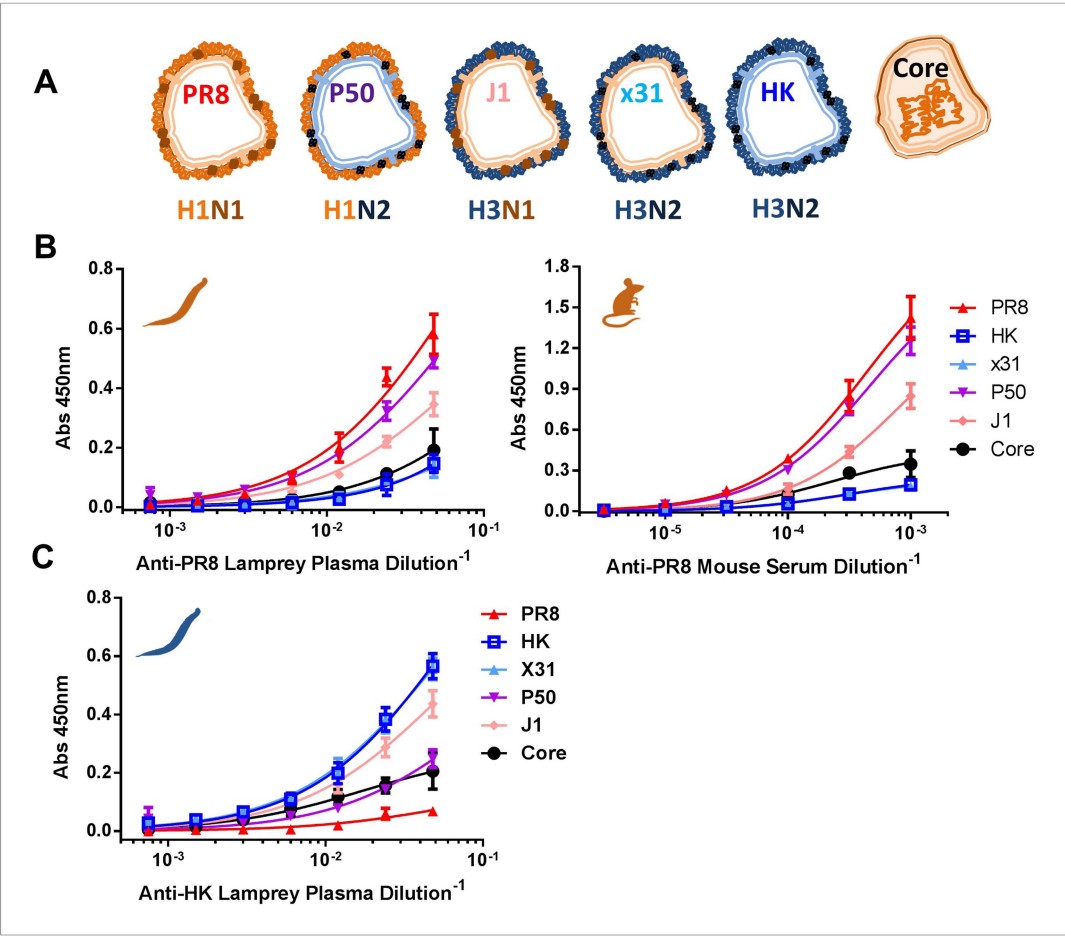

Figure 3. Immunodominance hierarchy against IAV for lamprey and mice is the same. (**A**) Scheme depicting reassortant virus components used for experiments in this figure. (**B**) Equal protein quantities split (HA/NA/M1) and core (NP/M1) antigens bound to ELISA plates were tested for binding to anti-PR8 mouse sera or lamprey plasma. Mouse data are representative of two mice with n = 4 independent experiments. Lamprey data are from three pooled animals with n = 4 independent experiments. (**C**) Same as *Figure 3B*, but using anti-HK lamprey plasma. Data are from three pooled animals with n = 4 ELISA replicates.

The following figure supplements are available for figure 3:

**Figure supplement 1**. Detergent-split reassorted viruses.

**Figure supplement 2**. PR8 antibodies (Abs) bind HA and NA but not M influenza proteins.

**Figure supplement 3**. PR8 immunized lamprey plasma binds purified NP protein, but not purified M1 by ELISA.

lamprey plasma gave HI titers of 1:30 against PR8, but <1:5 against an H3N2 IAV and B/Lee, an influenza B virus, which is serologically totally distinct from IAV (*Figure 4A*). Immune lamprey plasma also significantly inhibited PR8 infectivity in MDCK cells relative to naïve plasma (*Figure 4B*).

The vast majority of Igs that inhibit IAV hemagglutination and viral infectivity bind the HA globular domain. To test if this is also the major target of lamprey VLRBs, we used a panel of PR8 viruses with 3, 6, 9, or 12 amino acid substitutions located among the five defined antigenic sites (*Das et al., 2013*). ELISAs using intact wild-type or mutant viruses as immunoadsorbents show that lamprey plasma similarly detect antigenic drift in the globular domain, with a significant loss of binding with six substitutions and a loss of ~60% of binding with 12 substitutions (*Table 1*). Similar binding is seen with mouse, guinea pig, and chicken PR8 immune seras (*Table 1—source data 1*). Factoring in the

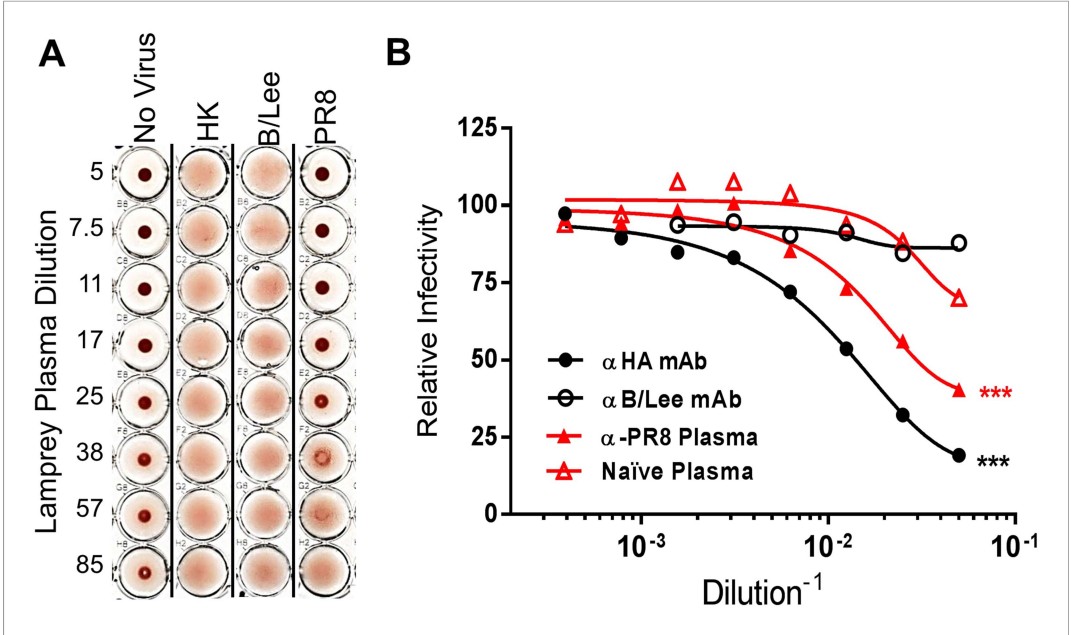

**Figure 4**. Lamprey VLRBs bind to hemagglutinin and neutralize infection. (**A**) Plasma from PR8-immunized lamprey inhibits PR8 hemagglutination at a 1:30 plasma dilution, but did not inhibit hemagglutination by either HK or B/Lee at any dilution. Data are representative of two experiments. (**B**) MDCK cells were infected with an MOI 0.07 of PR8 in the presence of titrated mAb supernatants (H17L2 against PR8 or control 1.2F4 against influenza B/Lee) or lamprey plasma (L9 vs Naïve). After 8 hr cells were fixed, double-stained with anti-HA and anti-NP Igs. Cells positive for either HA or NP by flow cytometry were considered infected. Data from four independent experiments were normalized to control for different percentages of infection between experiments and fit to a variable dose–response curve. The best-fit, calculated infectious dose 50 (ID50) was significantly lower for both the immunized plasma and PR8 specific Ig (***p < 0.001).

contribution of anti-NA Abs to the signal, this indicates that the bulk of mouse, guinea pig, chicken and lamprey HA-specific Ig and VLRB recognize the globular head domain, and therefore that the globular domain is the immunodominant antigen for both mice and lamprey. As predicted, anti-PR8 VLRBs could easily distinguish antigenic drift in H1N1 isolates from the 1940s and 1950s (*Table 1* and *Table 1—source data 2*).

The loss of lamprey VLRB and mouse Ig binding in similar proportions to the PR8 HA-head domain mutants implies that each recognizes similar epitopes and is subject to similar physicochemical rules of binding. To compare Ig and VLRB footprints, we competed lamprey plasma against mouse monoclonal Abs (mAbs) specific for defined HA antigenic sites for binding to PR8-coated ELISA wells. Although IAV immunization elicited VLRB responses in all lampreys tested, we selected lampreys with the highest titers for the competition experiments because these experiments require larger amounts of VLRB. Comparing the relative titers of inhibitory activity provides an approximate measure of binding proximity. Such competition assays are complicated by steric hindrance between Abs binding physically adjacent epitopes as well as more subtle positive and negative conformational effects that occur upon Ig binding to HA (*Lubeck and Gerhard, 1982*).

Anti-PR8 plasma from lampreys 7 and 9 (L7, L9) competed with the mAbs tested, whereas neither naïve lamprey plasma nor mAbs to an irrelevant antigen competed with the mAbs (*Table 2* and *Table 2—source data 1*). Also, while both L7 and L9 have a similar ELISA titer to whole virus, L9 plasma competes with mAbs specific for each of the five sites, but L7 fails to compete with Sa and Sb mAbs. This demonstrates that VLRB binding to the HA head does not uniformly block binding of all head-specific Igs and, importantly, indicates that fine antigen specificity varies among individual lampreys. We also infer this from the different titers observed against the various mAbs, an effect that is unlikely to be based on mAb affinity, since VLRBs are allowed to bind to HA prior to mAb addition.

**Table 1**. Lamprey plasma binding is sensitive to drifted viruses by ELISA and HI

| Virus | ELISA % ∆AUC* | HI Titer† |
|---|---|---|
| PR8 | – | 40 |
| # Substitutions in HA head | | |
| 3 | 15 | 10 |
| 6 | −25 | 20 |
| 9 | −46 | <10 |
| 12 | −59 | <10 |
| H1N1 isolates | | |
| A/Weiss/43 | −61 | – |
| A/Cameron/46 | −50 | – |
| A/Malaysia/54 | −77 | – |

*Equal amounts of each virus were plated and probed with lamprey plasma (L25) or mouse sera. Data from four independent experiments were normalized to the $B_{Max}$ of the stem binding mAb 3A01 to allow precise comparison between viruses and replicates (**Table 1—source data 3**). Data were fit to a hyperbola and the percent change in area under curve (AUC) between PR8 and the indicated virus is reported. All curve-means were significantly different from PR8 by Two Way ANOVA followed by Tukey Multiple comparisons test, p < 0.001, except PR8 vs 3.
†Agglutination inhibition of four HAU of input virus by lamprey plasma (L27) occurred at the dilution reported. HA, hemagglutinin.

**Source data 1**. Other animals show similar binding to sequential virus series. ELISA binding curves for lamprey plasma from **Table 1** are plotted alongside PR8 immunized guinea pig, chicken and mouse sera against the same plated Sequential virus series. Each graph shows representative data on a single animal's sera. Percent change in area under curve between wt PR8 and Sequential 12 is shown on each graph (∆AUC).

**Source data 2**. More H1N1 isolates. ELISA binding curves for lamprey plasma from **Table 1** against plated H1N1 isolates are plotted along with two additional isolates omitted from the Table.

**Source data 3**. Anti-HA stem Ab binding curves used to normalize amounts of plated HA in **Table 1**. (**A**) ELISA binding curves for serially diluted anti-HA stem Ab (3A01) added to the Sequential virus series are show from each of the four experiments. (**B**) Same as (**A**) for the H1N1 isolate panel.

To minimize steric effects, we extended these findings using Fab mAb fragment (25 kDa vs 150 kDa for intact Ig). The patterns observed with L7 and L9 were highly similar to those obtained with intact Igs. Plasma from an additional lamprey (L29) also effectively blocked each of the four Fabs tested (**Table 2** and **Table 2—source data 2**). L29 plasma did not, however, block binding of either a representative mouse or human mAb specific for the stem region with broadly neutralizing activity, suggesting that, as in mammals, the stem region is poorly immunogenic in lampreys (**Table 2** and **Table 2—source data 3**). Rather, as in man and mouse, the lamprey immune system focuses on the globular domain, and further, recognizes highly similar epitopes.

## Discussion

Our findings support the conclusion that Ig and VLRB demonstrate similar specificity for HA head epitopes. Although steric issues limit interpretation of the competition experiments, the common effect of amino acid substitution on VLRB and Ig binding argues strongly for convergent recognition of highly overlapping epitopes. Equally surprising is the similarity of the overall immunodominance profile of IAV immunogens in mammals and lamprey responding to inactivated intact virus: the HA globular domain is favored over the stem; HA is favored over NA; and internal proteins, despite their relative abundance in the virion, are weakly (NP) or undetectably (M1) immunogenic.

Our findings extend understanding of the cyclostome VLRB response. We confirmed the Pancer group's finding that total plasma VLRB concentration increases after immunization (**Alder et al., 2005**). Total Ig can also increase in jawed fish after immunization (**Castro et al., 2013**; **Xu et al., 2013**). This differs from mammals, where high levels are maintained constitutively. The ability of lampreys to respond to purified IAV without additional adjuvant indicates that lampreys recognize viral RNA or other viral innate immune activating molecules. We also show in competition assays that individual lampreys mount VLRB responses against different antigenic sites. Whether this is due to genetic, environmental, or stochastic factors is a question for further study. It is well worth noting that there is little information regarding how individuals among outbred groups vary in their Ig response to viruses or other complex antigens.

It is surprising that such structurally disparate molecules as Ig and VLRB receptors recognize the same proteins with overlapping epitopes. These recognition systems have evolved independently for >500 Mya, presumably shaped by different selective pressures from the environment, self-tolerance, and microorganisms. These differences notwithstanding, the four available VLRB-antigen crystal structures reveal many similarities in Ig and VLRB antigen recognition.

**Table 2**. IC$_{75}$ values for anti-PR8 lamprey plasmas and guinea pig serum (positive control) in competition with defined HA mAbs by ELISA*

| | L7 | | L9 | | L29 | | Guinea pig | Stem |
|---|---|---|---|---|---|---|---|---|
| Epitope† | IgG | Fab | IgG | Fab | IgG | Fab | IgG | C179 |
| Sa | NC‡ | NC | 250 | 400 | — | 290 | 36,000 | NC |
| Sb | NC | NC | 320 | 410 | — | 530 | 17,000 | — |
| Cb | 280 | 610 | 600 | 980 | — | 860 | 26,000 | — |
| Ca1 | 250 | 350 | 360 | 300 | — | 860 | 40,000 | — |
| Ca2 | 370 | — | 600 | — | — | — | 25,000 | — |
| Stem | — | — | — | — | NC | — | — | 3.5 |
| 2G02 | – | – | – | – | – | – | – | nM |

*Data was fit to a Hill Slope. IC$_{75}$ value was calculated from the curve using PRISM. IgG data are from three independent experiments, Fab data are from one experiment due to limited lamprey plasma.
†mAbs used—Sa: PEG-1; Sb: H28E23; Cb: H36 C12 (IgG), H9 D3 (Fab); Ca1: H2 4B1; Ca2: H18 S413; Stem: C179 and 2G02.
‡NC, no competition; '—', not determined.
mAbs, monoclonal antibodies.

**Source data 1**. Competition ELISA against α-Head HA panel Abs. Data from *Table 2* shown in graph form. Serially diluted unlabeled lamprey plasma raised against PR8 (L7, L9 or Naïve) was added to PR8 immobilized on 96 well ELISA plates. After 1 hr incubation, a fixed concentration of each indicated hybridoma supernatant (PEG-1, H28E23, H18 S413, H35 C12, H2 4B1, H18 S210, and Y8 2D1) was added at a predetermined concentration—65% of maximum binding (EC65). Data from three independent experiments were analyzed by Two Way ANOVA followed by Bonferroni Multiple Comparisons against the Naïve plasma data using PRISM. (*p < 0.05; **p < 0.01; ***p < 0.001; ****p < 0.0001).

**Source data 2**. Competition ELISA against α-Head HA Fabs. Same as *Table 2—source data 1* but with Fabs instead of hybridoma supernatants. p-value measurements determined with One-Way Anova followed by Dunnett's Multiple Comparison Test against Naïve plasma values. Stars indicate differences among whole groups. Data collected from only one experiment due to shortage of lamprey plasma.

**Source data 3**. Immune lamprey plasma does not compete against stem binding Abs by ELISA. Serially diluted naïve or immune lamprey plasma raised against PR8 (L29) on 96 well ELISA plates immobilized with PR8. After 1 hr incubation, a fixed concentration of purified monoclonal C179 or 2G02 was added at EC65. As a positive control, the two stem Abs were competed against each other or against an anti-HA head Ab (H28E23). Data are from at least two separate experiments with four total replicates. There was no statistical difference between the lamprey plasma curves. ELISA signal from these Abs is low, thus the curves are noisy. In contrast, the '2G02 then C179 curve' is statistically different from the 'H28E23 then 2G02 curve' by two-tailed t-test (**p< 0.01).

In the two VLRB structures with protein antigens (hen egg lysozyme and anthrax coat protein, BclA), the contact area ($\sim$1500 Å$^2$) is in the same range as reported for Igs (1400–2300 Å$^2$), and utilizes the same non-covalent forces (salt bridges, hydrogen bonds, and van der Waals contacts) in similar proportions to mediate binding, which requires close shape complementarity to the antigen (*Velikovsky et al., 2009*; *Kirchdoerfer et al., 2012*; *Deng et al., 2013*). In the VLRB-HEL structure, the relatively small, crescent-shaped VLRB binds to an epitope in the catalytic cleft, whereas larger, dimeric Ig VHVL Abs bind to flatter epitopes away from the catalytic site. Interestingly, structures of single-chain camelid VHH and shark IgNAR have revealed that they also favor the catalytic cleft of HEL that is presumably sterically inaccessible to dimeric VHVL Abs (*Velikovsky et al., 2009*).

Analysis of hundreds of VLRB sequences has previously revealed a bias towards aromatic amino acids at the variable positions on the concave surface (*Velikovsky et al., 2009*). In these analyses, less than half of the variable position residues contact antigen. When only the antigen *contacting* residue frequency is quantified, the amino acids are biased towards Tyr, Trp, Asn, and Asp residues (*Figure 5*). A similar bias towards these residues in antigen-contacting positions of Ig has also been observed

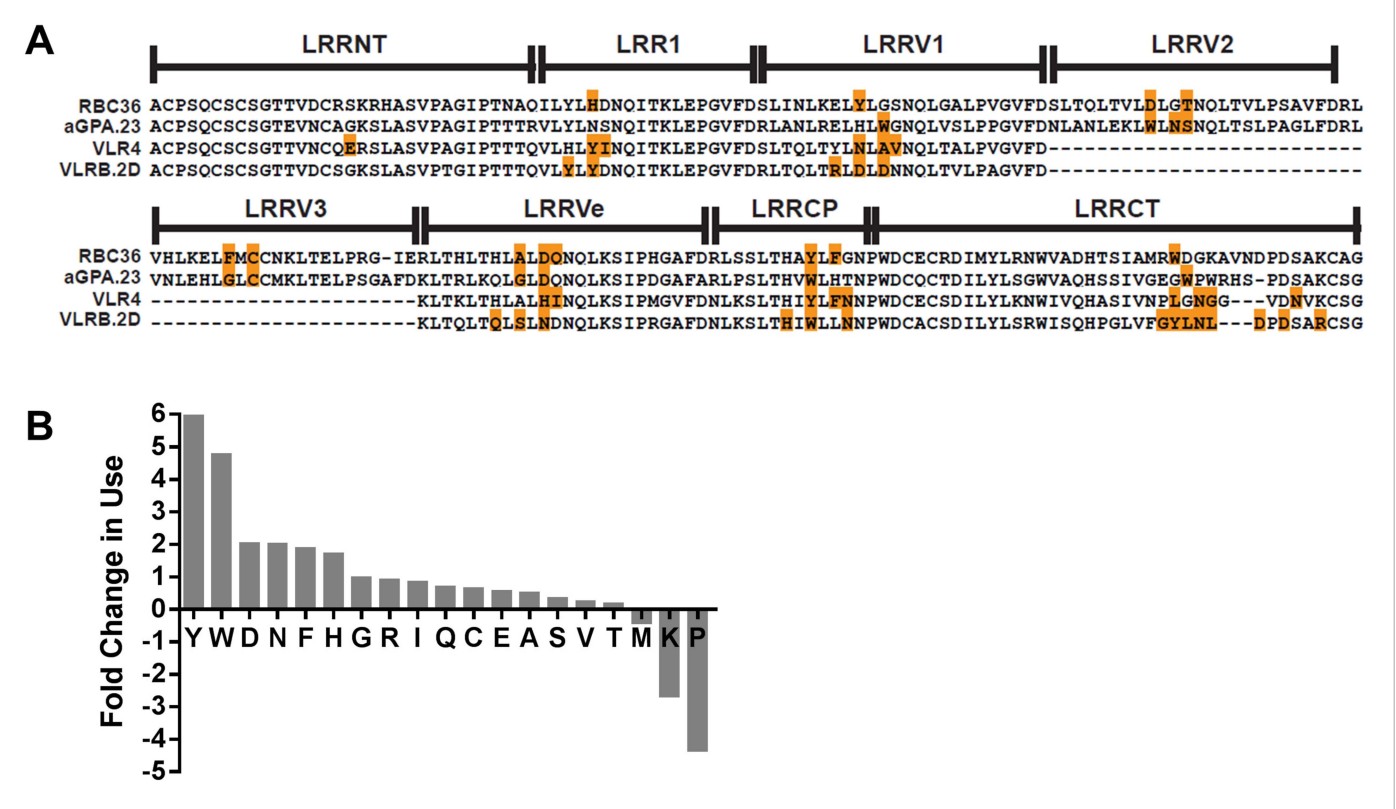

**Figure 5**. Paratope signature of VLRBs. (**A**) Contact residues determined by the crystal structures of VLRBs in complex with their antigens are highlighted in orange. RBC36 against H trisaccharide (3E6J); aGPA.23 against TF disaccharide (4K79); VLR4 against BclA (3TWI); and VLRB.2D against HEL (3G39). (**B**) Enrichment or shortfall of each amino acid in the contact residues relative to the total amino acids found in the full VLRB was determined from the ratio of frequency of each amino acid in contact residues vs the frequency in the total VLRB sequence. Leucine was excluded from the analysis as it is the major structural amino acid of VLRBs. No M, K, or P were found among the contact residues. Shortfall was determined by estimating, based on total VLRB frequency, how many amino acids would be there if the amino acid distribution was even throughout the VLRB.

(*Mian et al., 1991*; *Davies and Cohen, 1996*; *Ramaraj et al., 2012*; *Robin et al., 2014*). These similarities may account for the recognition of similar epitopes. If so, this may also represent the optimal general solution for producing a single family of receptors capable of recognizing what is essentially an infinite array of antigens with high specificity and affinity.

The universality of Ig and VLRB antigenicity and immunogenicity illuminated by our findings provides strong support for the utility of animal models for understanding human Ab responses to vaccines and other medically relevant immunogens. Perhaps, when it comes to Ab responses, neither mice nor lamprey lie, after all.

## Materials and methods

### Antibodies

We used the following Abs for ELISA and immunoblot experiments: 1:3000 mouse α-HA1 mAb, CM1 (*Magadan et al., 2013*); 1:2000 α-M1, M2-1C6, anti-Mouse recognizes 9 kDa N-terminal fragment (*Yewdell et al., 1981*); 1:3000 α-NP C-Terminal rabbit polyclonal 2364, 487–498; 1:10,000 α-NP HB-65 (*Yewdell et al., 1981*); 1:3000 α-NA C-Terminal anti-Rabbit polyclonal (*Dolan et al., 2010*); 1:2500 mouse α-lamprey VLRB 4C4; 1:100 C179 α-HA Stem anti-Mouse mAb (Takara); α-HA stem anti Human 3A01 and SFV005 2G02 g02; 1:5000 Donkey α-Mouse IRDye 800 nm (Li-Cor); 1:5000 Donkey anti-Rabbit IRDye 680 nm (Li-Cor); 1:5000 Mouse α-Flag M2 (Sigma); 1:2000 α-Mouse Kappa-HRP (Southern Bioscience); 1:2000 α-Rabbit Kappa-HRP (Jackson); 1:2000 α-Human-HRP (Jackson); and 1:2000 α-Guinea Pig-HRP (Jackson).

## Virus preparation and purification

Viruses B/Lee, A/HK/68 (HK), A/PR8/MCa (PR8), X31 (HK HA and NA, PR8 background), J1 (HK HA, PR8 background), and P50 (PR8 HA, HK background) were grown by pipetting 250 TCID50 viral units diluted in 50 µl 0.1% BSS/BSA into 10-day-old eggs. We collected allantoic fluid after 48 hr, clarified at 3000 RPM for 20 min, and pelleted virus through 20% sucrose by centrifuging for 2 hr at 26,000×$g$. We incubated pellets overnight in 2 ml PBS with calcium and magnesium (PBS++) and purified by centrifuging virus on a discontinuous 15–60% sucrose gradient, collecting virus at the interface, and pelleting 34,000×$g$ for 2 hr. After resuspending the pellet in 500 µl PBS++ overnight, we completely inactivated viral infectivity (as determined by adding to MDCK cells) by exposing for 20 min to 254 nm light at 2.4 mW/cm$^2$, and sterilized virus by passing through a 0.22-µm PDVF filter (Millipore). We measured total viral protein with the DC Protein Assay (Bio-Rad). We fractionated purified viruses by incubating 250 µl purified virus with 250 µl 15% octyl-β-glucoside. After pipetting until the opalescent solution became clear, we added 50 µl 10% NP-40and pipetted further before adding 950 µl PBS++. We centrifuged virus at 50,000×$g$ for 2 hr at 4°C, collected the top 1 ml and removed detergents with detergent removal spin column (Pierce). We resuspended the pellet (cores) in 1 ml PBS++, repelleted and sonicated cores into PBS ++.

## Immunizations

Lamprey larvae (*Petromyzon marinus*) captured from the wild by commercial fishermen (Lamprey Services, MI) were housed in sand-lined aquariums maintained at 16–18°C using a water chiller and fed brewer's yeast. The lampreys were immunized three times by intracoelomic cavity injections spaced 2–3 weeks apart containing ~10 µg virus diluted in 30 µl of 0.67× PBS (to match lamprey tonicity). 2 weeks after final immunization, ~200 µl lamprey plasma was collected in 300 µl of 30 mM EDTA, an anticoagulant and stored at 4°C in 20 mM MOPS pH 7.2 buffer and 0.025% sodium azide to prevent microbial growth. Plasma was also collected from non-immunized lampreys to serve as naïve controls. Leukocytes were harvested from the blood of each animal by collecting cells at the interface of a 55% Percoll gradient and banked in RNAlater (Qiagen) for future characterization. Guinea pigs and mice were immunized with an intramuscular injection of ~10 µg virus in 25–50 µl PBS and boosted 2 weeks later. Serum was collected 2 weeks after the boost. Mouse and guinea pig studies were approved by and performed in accordance with the Animal Care and Use Committee of the National Institute of Allergy and Infectious Diseases.

## ELISAs

We incubated purified IAV (~0.05 µg per well in 100 µl PBS++), purified A/WSN/33 NP (*Ye et al., 2013*) generously provided by Dr Yizhi Jane Tao and Dr Yukimatsu Toh at Rice University (100 nmol/well), or purified matrix protein (*Oxford and Schild, 1976*) (4 nmol/well) in Immulon 4HBX 96 well plates for 12 hr to 7 days at 4°C on an orbital shaker (all incubations were similarly shaken). Just before using, we washed plates 3× with PBS + 0.05% Tween-20 (PBST). We then incubated 100 µl diluted primary sera for 1 hr at 4°C, washed with PBST, and for lamprey plasma, incubated with 100 µl 4.4 nM mouse anti-VLR 4C4 mAb for 1 hr at 4°C. After washing 3× with PBST, we incubated wells with 100 µl 1:2000 HRP anti-mouse κ chain (Southern Biotech) in PBS++ 1 hr at 4°C, washed 3× with PBST and added 100 µl/well SureBlue Peroxidase Substrate (KPL). After 5 min at room temperature, we inactivated HRP with 50 µl/well 1 M HCl. And measured absorbance at 450 nm. The absorbance data was graphed and fit to Hill Equation using PRISM software. For competition ELISA, we first incubated wells with competing Ig or VLRB for 1 hr at 4°C, and then added mouse mAbs as supernatants or purified Fab fragments at a concentration equivalent to the Igs' EC$_{65}$ binding. Data was fit to a Hill Equation from which the IC$_{75}$ was calculated in PRISM. After an hour at 4°C, we washed the plates, and developed with anti-mouse HRP Ig and peroxidase substrate, as described above.

## Protein gels and immunoblotting

We mixed purified IAV (0.5–4 µg protein) with 4× NuPage loading buffer (Invitrogen), with or without 4 mM DTT, and boiled for 15 min at 96°C. We electrophoresed samples with SeeBlue Plus2 ladder on 4–12% Bis-Tris Gels (Invitrogen) at 180 V for 90 min. To visualize proteins, we fixed gels for 10 min with 10 ml 10% acetic acid and 50% methanol, shaking at RT. After removing fixative we added 10 ml GelCode stain (Pierce) and shook for 30 min at room temperature, then destained the gels with water

overnight. For immunoblotting, we transferred proteins from gels to PVDF membranes with the iBLOT at P3 setting for 7 min. We blocked membranes for 1 hr at room temperature with either 10% BSA in water for blots probed with lamprey plasma or with StartingBlock for mouse Abs (Thermo). After incubating with primary Ig or VLRB 1 hr at room temperature, washing 5× for 5 min each in TBST (10 mM Tris, 150 mM NaCl, 0.1% Tween-20), we added secondary and tertiary Ig, repeating the washing step after each incubation. We imaged blots on a Li-Cor Odyssey.

## Plasmid transfection and flow cytometry

We transfected HeLa cells using Lipofectamine LTX (Life Technologies) with PR8 proteins HA, NA, M, NP, or NS1 in a pDZ vector and cultured for 24 hr. To enable Ig or VLRB access to internal proteins, we fixed and permeabilized NP, NS1, and M transfected cells with FoxP3 buffer (eBiosciences). We stained all cells with appropriate dye-labeled Ig, mouse sera, or lamprey plasma and analyzed samples using a LSR II flow cytometer and fitted data to a one-site binding hyperbola model using PRISM software.

## Ab functional inhibition assays

For HI, we treated lamprey plasma for 56°C for 30 min and incubated with four HAU PR8 before adding human O+ erythrocytes. For neutralization, we cultured MDCK cells (350k) in 24-well plates overnight. The next day we added PR8 at a MOI of 0.07 in the presence of either H17-L2 (anti HA mAb) or control 1.2F4 against influenza B, or lamprey plasma, animal L9 vs naïve plasma diluted in 0.1% BSS/BSA. After 1 hr incubation, we removed the supernatant and replaced with complete media. After 7 more hours, we trypsinized cells, fixed and permeabilized cells with FoxP3 Buffer and stained with anti-HA and anti-NP mAbs, then analyzed cells with a LSR II flow cytometer. Single cells positive for either HA or NP by flow cytometry were considered infected. Data were fitted to a variable dose-response curve and the best-fit infectious dose 50 (ID50) calculated using PRISM software.

## Acknowledgements

The authors thank Davide Angeletti, Christopher Brooke, Greg Frank, William Ince, and James Gibbs for critical reagents and ideas.

## Additional information

### Funding

| Funder | Grant reference | Author |
| --- | --- | --- |
| Division of Intramural Research, National Institute of Allergy and Infectious Diseases | | Meghan O Altman, Jack R Bennink, Jonathan W Yewdell |
| National Institutes of Health | R01 AI072435 | Brantley R Herrin |
| Georgia Research Alliance | R01 GM100151 | Brantley R Herrin |

The funders had no role in study design, data collection and interpretation, or the decision to submit the work for publication.

### Author contributions

MOA, BRH, Conception and design, Acquisition of data, Analysis and interpretation of data, Drafting or revising the article, Contributed unpublished essential data or reagents; JRB, JWY, Conception and design, Analysis and interpretation of data, Drafting or revising the article, Contributed unpublished essential data or reagents

### Ethics

Animal experimentation: This study was performed in strict accordance with the recommendations in the Guide for the Care and Use of Laboratory Animals of the National Institutes of Health.

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
