## [Decision Letter]

Thank you for sending your work entitled “Lamprey VLRB Response to Influenza Virus Supports Universal Rules of Immunogenicity and Antigenicity” for consideration at *eLife*. Your article has been favorably evaluated by Tadatsugu Taniguchi (Senior Editor), a Guest Reviewing Editor, and two reviewers.

The Reviewing Editor and the reviewers discussed their comments before reaching this decision, and the Reviewing Editor has assembled the following comments to help you prepare a revised submission.

The reviewers agree that these results strongly support the idea that protein epitopes share certain physicochemical properties, in addition to simple exposure on the antigen surface, which renders them immunogenic in both jawed and jawless vertebrates.

Major concern:

It must be noted that the paper´s conclusion is not entirely original. Specifically, Altman et al. should refer to Figures 1 and 3 in [28]. In that study, a comparison of crystal structures revealed that the epitope on hen egg lysozyme recognized by a lamprey VLRB almost completely overlapped the epitope recognized by anti-lysozyme VH antibodies from camel and shark. The same study also reported that Trp and Tyr are the two residues which occur at highest frequency in the ligand-contacting of both VLRBs and Igs, as also seen in Figure 5 of the present work. The authors should revise their text accordingly.

Minor comments:

1) Lampreys are poikilotherms. Temperature can strongly impact behavior in response to infection and also influences the adaptive immune response in bony fish. In addition, temperature has an impact on the stability of proteins and low temperature can stabilize exposed flexible loops. The authors should describe the environment in which the lampreys were kept, most importantly the temperature of water. In addition, the authors could mention this possible effect of temperature on VLRB responses in lampreys in the Discussion.

2) Mouse sera bound to permeabilized pDZ-NP transfected Hela cells and lamprey plasma did not. This is an interesting difference. However, both mouse serum and lamprey larvae bound to virions cores in ELISA. This virion core is composed of NP and likely the vRNPs. Are there lamprey VLRBs against the vRNPs and no IgG against vRNPs in the mouse serum? This could be tested using cells transfected with polymerase expression plasmids. Alternatively, NP-specific antibody or VLRB responses may depend more on antigen processing and presentation of NP in the virion than HA and NA specific responses. How similar are these processes in lamprey and mice? This possibility could be mentioned in the Discussion.

There could also be a technical explanation for the lack of binding of VLRBs to NP in transfected cells. Can the authors exclude that VLRBs cannot enter permeabilized cells?

3) The lack of competition of lamprey immune serum against HA stem specific mabs is based on a negative result (Table 2). A positive control for competition should be included. Presumably C179 and human 2G02 should compete against each other.

---

## [Author Response]

*It must be noted that the paper´s conclusion is not entirely original. Specifically, Altman et al. should refer to*
Figures 1 and 3
*in*
[28]*. In that study, a comparison of crystal structures revealed that the epitope on hen egg lysozyme recognized by a lamprey VLRB almost completely overlapped the epitope recognized by anti-lysozyme VH antibodies from camel and shark. The same study also reported that Trp and Tyr are the two residues which occur at highest frequency in the ligand-contacting of both VLRBs and Igs, as also seen in*
Figure 5
*of the present work. The authors should revise their text accordingly*.

Velikovsky et al. made a number of observations relevant to the present study, and we have modified the Discussion to better acknowledge their important contribution. However, there are several notable differences between the two studies.

Velikovsky et al. described a single VLR from a lamprey immunized with a single protein antigen. Obviously, while the structural description of the VLR antigen interaction is elegant, it is not possible to draw general conclusions about how VLRs interact with protein antigens compared with immunoglobulins. The major conclusion of our paper is that the rules of immunogenicity for a complex viral antigen at the level of individual proteins and epitopes is conserved between mammals and lampreys.

The preference for Tyr and Trp in “ligand-contacting” residues in Velikovsky et al. was based on the alignment of 588 VLRB sequences and quantifying the residues present at the variable positions of each LRR β-strand (residues at the x position in the xLxLxx motif). Although phrased “ligand-contacting” in by Velikovsky et al., the analysis was based on total variable-position residues rather than predicting residues that actually contact antigen based on crystal structure data. The distinction is subtle yet significant, because only roughly one-half (VLRB.2D, 11 of 20; VLR4, 11 of 20) to one-third (RBC36, 11 of 28; aGPA.23, 9 of 28) of the residues in the variable positions actually contact antigen in the available structures. For instance, Asp is the most frequent “ligand-contacting” residue in VLRB.2D, occupying 5 of 20 total positions, yet only two Asp residues actually contact HEL making it fourth most frequent in our analysis (Velikovsky et al., Figure 5, red circles above entropy bars).

*Minor comments*:

*1) Lampreys are poikilotherms. Temperature can strongly impact behavior in response to infection and also influences the adaptive immune response in bony fish. In addition, temperature has an impact on the stability of proteins and low temperature can stabilize exposed flexible loops. The authors should describe the environment in which the lampreys were kept, most importantly the temperature of water. In addition, the authors could mention this possible effect of temperature on VLRB responses in lampreys in the Discussion*.

The lamprey aquaculture facility is maintained at an ambient air temperature of 16-18^o^C, and we have also housed the animals at room temperature (∼23^o^C) for some experiments. Although we have not rigorously examined the effect of ambient temperature, we haven’t noted obvious temperature related difference in the magnitude or kinetics of the VLRB. The Methods section has been modified to include the aquaculture conditions and temperature information. Since the VLR responses are comparable at the two aquaculture temperatures we have tested, we do not believe that this will add to the Discussion.

*2) Mouse sera bound to permeabilized pDZ-NP transfected Hela cells and lamprey plasma did not. This is an interesting difference. However, both mouse serum and lamprey larvae bound to virions cores in Elisa. This virion core is composed of NP and likely the vRNPs. Are there lamprey VLRBs against the vRNPs and no IgG against vRNPs in the mouse serum? This could be tested using cells transfected with polymerase expression plasmids. Alternatively, NP-specific antibody or VLRB responses may depend more on antigen processing and presentation of NP in the virion than HA and NA specific responses. How similar are these processes in lamprey and mice? This possibility could be mentioned in the Discussion*.

There could also be a technical explanation for the lack of binding of VLRBs to NP in transfected cells. Can the authors exclude that VLRBs cannot enter permeabilized cells?

The discrepancy between lamprey plasma binding viral cores, but not NP in a flow-based pDZ-NP assay puzzled us as well. We predicted anti-NP VLRBs would be in the immune plasma. To directly address this discrepancy and address the reviewers concerns, we obtained NP generously provided by Yizhe Jane Tao’s lab at Rice University that was sufficiently pure for use in x-ray crystallography studies. In contrast to the flow assay, both immune mouse sera and lamprey plasma bound purified NP by ELISA, clearly demonstrating that lamprey, like mice, respond to NP. These anti-NP VLRBs are likely to account for most of the binding to flu core antigens. As predicted by the reviewers, the lack of VLRB binding to NP expressing cells in the flow assay most likely has a technical explanation.

We also probed purified M1 protein by ELISA, and as expected, neither the mouse sera nor lamprey mounted a detectable antibody response. Virions only have ∼10 copies of the polymerase complex (∼1% of the major structural proteins. This is at or below the limit of detection of the ELISA we use.

*3) The lack of competition of lamprey immune serum against HA stem specific mabs is based on a negative result (*Table 2*). A positive control for competition should be included. Presumably C179 and human 2G02 should compete against each other*.

This is an excellent suggestion. Mouse C179 does indeed compete with 2G02 by competition ELISA. The requested control has been added to Table 2, and the full curves have been added to [Supplementary-material SD6-data].